# From Machine to Human Learning: Towards Warm-Starting Teacher Algorithms with Reinforcement Learning Agents

## Abstract

We present an investigation into using Reinforcement Learning (RL) agents to address the well-established cold-start problem in AI teacher algorithms that require extensive human learning data. While the challenge of bootstrapping personalized learning systems is recognized across domains, collecting comprehensive human learning data remains resource-intensive and often impractical. Our work explores a novel methodological approach: warm-starting data-hungry teacher algorithms using RL agents to provide an initial foundation that can be refined and augmented with human learning data. We emphasize that this approach is not intended to replace human data, but rather to provide a practical starting point when such data is scarce. Through exploratory experiments in two game-based environments—a Super Mario-inspired platformer and an Overcooked-inspired medical training simulation—we conduct human subjects studies demonstrating that RL-initialized curricula can achieve comparable performance to expert-crafted sequences. Our preliminary analysis reveals that while human learning outcomes are positive, there remain notable gaps between RL agent behavior and human learning patterns, highlighting opportunities for improved alignment. This work establishes a promising potential for RL-initialized teaching systems, opening valuable research directions at the intersection of RL and human learning.

## 1 Introduction

Artificial Intelligence (AI) applications in education hold the promise of revolutionizing learning through scalable, personalized, and adaptive approaches (Doroudi et al., 2019; Alrakhawi et al., 2023). These AI-driven methods aim to address the limitations of traditional expert-designed curricula, which often struggle to efficiently meet the diverse needs of a vast and growing student population across an expanding knowledge base (Lin et al., 2023). In theory, AI tools could simultaneously provide tailored learning experiences to numerous students, dynamically adapting to individual needs and learning styles (Mousavinasab et al., 2021). However, recent studies have shown that learning-based teacher algorithms often underperform when compared to expert-initialized or even random algorithms (Green et al., 2011; Lindsey et al., 2014).

These systems require extensive data on student's learning process in order to design effective curricula (van der Velde et al., 2024; Doroudi et al., 2019). However, gathering comprehensive human learning data is time-consuming and costly; in one study, it took approximately 900 man-hours for a Machine Learning-based teacher algorithm to converge (Bassen et al., 2020). While existing approaches supplement human data by incorporating demographic information (Zhao et al., 2020; Patel & Thakkar, 2022), this method introduces potential biases and privacy concerns (Suresh et al., 2022; Wang et al., 2018), limiting the development of robust teaching strategies. The challenge is especially significant in dynamic fields where learning patterns change rapidly, requiring constant data collection and algorithm updates (Hatzilygeroudis & Prentzas, 2004).

Our work focuses on teacher algorithms that adaptively sequence training tasks to optimize student learning outcomes. These algorithms interact with students by assigning targeted challenges, creating personalized curricula that evolve with student progress. Motivated by the capabilities of Reinforcement Learning (RL) agents in mastering complex environments (Silver et al., 2017;

2016), we propose leveraging these agents to bootstrap training data for teacher algorithms. This novel methodological approach aims to augment early algorithm development, reducing initial data requirements while providing a foundation that can be refined with human learning patterns. We evaluate this approach through human subjects studies in two contrasting environments: a Super Mario-style platformer for motor skills and a medical emergency response simulation with discrete tasks. Our findings suggest this approach offers a promising direction for addressing the cold-start problem in adaptive teaching systems. We invite the research community to explore advancing RL-based initialization with human learning patterns, potentially enabling more accessible personalized learning technologies.

Our key contributions are as follows:

1. We introduce a two-stage framework that leverages RL agents to generate training data for teacher algorithms that optimize student learning through task recommendations.

2. We present two pedagogy-based teacher algorithms under this framework: a human-friendly adaptation of PERM (Tio & Varakantham, 2023) for domains with potentially infinite scenarios, represented by a finite set of parameters; and SimMAC, a novel Task Sequencing algorithm for domains with a finite and discrete set of scenarios.

3. We demonstrate our approach's effectiveness through two new environments, the Jumper game and Emergency Response game, where human trials show our methods outperform baselines approaches and match expert-handcrafted curricula.

## 2 RELATED WORK

While our work focuses on human learning, it draws inspiration from the substantial body of research on curriculum learning in RL (Narvekar et al., 2020; Da Silva & Costa, 2019). These methods have demonstrated significant success in training artificial agents by automatically generating learning sequences that improve sample efficiency and convergence. However, existing curriculum learning approaches in RL are fundamentally designed around the affordances of artificial agents and are therefore unsuitable for human learners.

We identify three characteristics that current RL curriculum methods has that do not hold for human learning contexts. First, they assume access to massive amounts of training data. RL agents can collect millions of experiences through parallelized environments and accelerated simulation, while human learners operate on much smaller data scales with natural time constraints. Second, many approaches exploit artificial agent properties that have no human equivalent. For instance, direct access to value functions enables curriculum methods to use the learner's internal state for optimization (Parker-Holder et al., 2022), which is impossible with human learners. Similarly, Sukhbaatar et al. (2017) employs self-play to generate curricula, which would require training an artificial proxy (e.g. through Imitation Learning (Torabi et al., 2018)) of the human learner (leading back to our original cold-start problem). Third, these methods operate on timescales incompatible with human learning. ACCEL (Parker-Holder et al., 2022) uses evolutionary search over thousands of generations to discover useful training environments, while Matiisen et al. (2019) required 4500 training epochs to learn effective task-selection rules. Both approaches far exceed practical human study durations and would be too slow to provide useful adaptation during human learning sessions.

Recent research has explored using RL to optimize instructional activities in education (Doroudi et al., 2019). However, across different domains, data-hungry RL teachers have shown mixed results, often failing to outperform baselines (Green et al., 2011; Segal et al., 2018; Doroudi et al., 2017). A key challenge is the complexity of modeling student states, requiring an "inordinate amount of data" (Doroudi et al., 2019). Recent RL implementations in algebra education show promise but face challenges, notably the cold-start problem. (Bassen et al., 2020) reported their RL teacher needed nearly 600 learner course completions, or 900 man-hours, to converge on an effective strategy. This highlights a critical challenge in applying learning-based methods to human learning: the need for extensive initial data to achieve competency, raising practical and ethical concerns for real-world educational implementation. To address these issues, our study proposes employing RL agents as warm-start human learners for data collection. We aim to generate valuable training data for teacher algorithms, potentially mitigating the cold-start problem and improving the overall effectiveness of AI-assisted education.

We focus on two key principles to guide effective learning. First, both human (Van den Akker, 2007; Grant, 2018; Macalister & Nation, 2019) and artificial learners (Bengio et al., 2009; Graves et al., 2017; Huang et al., 2020) benefit from progressively challenging curricula, where task difficulty gradually increases to match student abilities. This alignment with the Zone of Proximal Development (Vygotsky & Cole, 1978) ensures optimal learning by maintaining an appropriate challenge level. Second, learning continuity enhances knowledge acquisition by connecting new content to prior experiences, creating smoother transitions through content overlap. This spiral curriculum approach (Bruner, 2009) strategically leverages existing knowledge while increasing difficulty, making learning more intuitive and effective than introducing entirely new content. Our proposed teacher algorithms address these principles: both incorporate difficulty progression, while SimMAC (Section 4.2) additionally considers task similarity by selecting subsequent tasks based on the learner's experience history.

## 3   TEACHER PROBLEM

We study interactive teaching where algorithms dynamically assign tasks based on student performance feedback to maximize learning outcomes. Our focus encompasses two paradigms: UED and Task Sequencing.

**Unsupervised Environment Design**   UED (Dennis et al., 2020) generates diverse challenges to optimize student learning. The core assumption is that exposing students to diverse environments fosters generalized proficiency across the environment distribution, enhancing generalization.

Formally, UED is conceptualized as an Underspecified Partially Observable Markov Decision Process (UPOMDP), defined as $\mathcal{M} = \langle A, O, \Theta, S, T, I, R, \gamma \rangle$, where $A$ represents the action space, $O$ the observation space, $S$ the state space, $T : S \times A \times \Theta \to \Delta(S)$ the transition function, $I : S \times \Theta \to \Delta(O)$ the observation function, $R : S \times A \times S \times \Theta \to \mathbb{R}$ the reward function, and $\gamma \in [0, 1)$ the discount factor. The UPOMDP extends the traditional POMDP by incorporating $\Theta$, a set of environment parameters where $\theta \in \Theta$ represents specific configurations that define task instances. At each timestep $t$, the teacher selects $\theta_t \in \Theta$ to generate an environment instance $\mathcal{T}^{\theta_t}$ with state $s_t \in S$, allowing dynamic adjustment of challenge complexity based on observed student performance. For example, in a navigation task, $\theta$ might parameterize obstacle frequency, enabling progressive difficulty calibration to maximize learning outcomes across $\Theta$.

**Task Sequencing**   Task Sequencing represents a constrained UPOMDP where $\Theta$ defines a discrete and finite task pool with varying difficulty levels and knowledge requirements, requiring agents to apply different knowledge sets for successful completion. A successful teacher would determine optimal task ordering to maximize learning efficiency and post-training generalization across the task distribution. Given its versatility and effectiveness, Task Sequencing finds widespread application in various educational contexts (Bassen et al., 2020; Segal et al., 2018).

## 4   RL-SUPPORTED TEACHER ALGORITHMS

In this section, we detail our two-stage process for using RL to retrieve data for our teacher algorithms, consisting of an *Exploration Stage* and an *Exploitation Stage*. We then present two algorithms that benefits from this process: PERM-H, a human-adapted version of existing work, and SimMAC, a novel approach specifically designed for Task Sequencing.

**The Exploration Stage**   In the first stage, we use RL agents to simulate student-environment interactions and collect data. These RL agents interact with a variety of levels generated using DR (Tobin et al., 2017). We record the agents' performance, the parameters of the levels they encounter, and other relevant data specific to the teacher algorithms we're developing. The key idea here is to use RL agents as stand-ins for human students. This allows us to gather extensive data on learning progress without requiring actual human participants. An important advantage of this approach is that RL agents start from scratch and improve over time, much like real students. This enables us to simulate a diverse group of learners with varying skill levels, providing a rich dataset for our teacher algorithms to learn from. By using RL agents in this way, we can generate a large amount

of valuable training data for our teacher algorithms, helping to address the cold-start problem and potentially improve the effectiveness of AI-assisted education from the outset.

**The Exploitation Stage**   In the exploitation stage, we utilize the data collected during the exploration stage to train the teacher algorithms and apply compatible algorithms to human training. Similar to RL training under UPOMDPs, we emulate the process with humans using a continuous loop. We note here that as more human interaction data is collected, it can be used to supplement, and eventually replace, RL data for stronger alignment to humans.

The teacher algorithm first makes an inference based on the student's recent performance $r_t$ and outputs the next task, $\theta_{t+1}$. The student then trains under the new level generated from $\theta_{t+1}$ and returns the corresponding reward or performance metric, $r_{t+1}$. This iterative process continues throughout the training session until a predetermined termination criterion is reached.

### 4.1   PERM-H

PERM (Tio & Varakantham, 2023) is an Item-Response Theory-based model for UED in RL that infers agent ability $a$ and environment difficulty $\delta$ from observed parameters and performance to determine subsequent training environments, motivated by the Zone of Proximal Development (Vygotsky & Cole, 1978). We modified PERM's original assumption that optimal learning occurs when $\delta = a$ to $\delta = \epsilon a$ ($\epsilon \geq 1.0$), accommodating potentially faster human learning rates (Tsividis et al., 2017). We call this adaptation PERM-H.

During the Exploration Stage, we collect $\theta$ and $r$ to train PERM-H. In the Exploitation stage, PERM-H operates cyclically by estimating the student's current ability, using this estimate to specify the desired difficulty for the next level, and generating a level matching this difficulty, while adapting to the student's progress. While effective for difficulty-based progression, PERM-H, without major modifications, cannot handle domains requiring distinct, non-comparable skills. For these cases, we developed an alternative algorithm for more diverse task sequencing.

### 4.2   SIMMAC

SimMAC creates effective learning curricula by balancing task difficulty and knowledge continuity. Our approach is built on two fundamental principles: tasks requiring less training time are inherently easier, and optimal learning occurs when new tasks build upon previously acquired knowledge. Silva & Costa (2018) employs similar principles of task similarity by constructing a graph based on objects present in the environments. We take a different approach by discovering similarities without requiring explicit environmental feature declarations.

**Quantifying Task Difficulty**   We measure task difficulty through convergence analysis: training an RL agent uniformly across tasks and identifying the point at which performance stabilizes. We consider task 1 easier than task 2 if and only if its convergence point $c_\theta$ occurs earlier ($c_{\theta_1} < c_{\theta_2}$). We average results across multiple runs to ensure measurement reliability.

**Modeling Knowledge Transfer Between Tasks**   The core innovation of SimMAC lies in its ability to identify knowledge overlap between tasks. We approximate a task's knowledge content through trajectory analysis, operating on the principle that similar tasks elicit similar behavioral patterns during solution.

A trajectory $\tau$ represents the sequence of states and actions, i.e., $\tau = \{s_0, a_0, s_1, a_1, ..., a_{T-1}, s_T\}$. The distribution of trajectories, the occupancy measure, provides a mathematical expression of the knowledge required for task completion:

$$\rho_{\mathcal{T}^\theta}^\pi(s, a) = \sum_{t=0}^{T} \Big[ Pr(s_t = s, a_t = a | s_0 \sim p_0(\cdot), s_t \sim$$

$$p(\cdot | s_{t-1}, a_{t-1}, \theta), a_t \sim \pi(\cdot | s_t) \Big]$$

where $T$ is the horizon limit, $p_0(\cdot)$ is the initial state distribution.

Tasks with overlapping occupancy measures require similar actions in similar states, indicating shared knowledge requirements. We quantify this similarity using Wasserstein distance $\mathcal{W}$ between trajectory distributions (Li et al., 2023b) $\mathcal{W}(\rho^{\pi}_{\mathcal{T}^{\theta_i}}, \rho^{\pi}_{\mathcal{T}^{\theta_j}}) \approx \mathcal{W}(\tau_i, \tau_j)$ where $\rho^{\pi}_{\mathcal{T}^{\theta_i}}$ and $\rho^{\pi}_{\mathcal{T}^{\theta_j}}$ represent the occupancy measures induced by policy $\pi$ on task $\mathcal{T}^{\theta_i}$ and task $\mathcal{T}^{\theta_j}$, respectively, with $\tau_i$ and $\tau_j$ being the resulting trajectories.

Extending beyond Li et al. (2023b)'s pairwise comparisons, we measure similarity between a candidate task and the entire set of previously completed tasks: $\mathcal{T}^{\theta_k}$ and a set of tasks, $\mathcal{T}^{\theta_{i\sim j}} = \left\{ \mathcal{T}^{\theta_i}, \mathcal{T}^{\theta_{i+1}}, ..., \mathcal{T}^{\theta_j} \right\}$. We aggregate the trajectories collected in $\mathcal{T}^{\theta_{i\sim j}}$ as $\tau_{i\sim j}$ and compute the distance $d$ between $\tau_k$ and $\tau_{i\sim j}$:

$$d(\mathcal{T}^{\theta_k}, \mathcal{T}^{\theta_{i\sim j}})\mathcal{W}(\rho^{\pi}_{\mathcal{T}^{\theta_k}}, \rho^{\pi}_{\mathcal{T}^{\theta_{i\sim j}}}) \approx \mathcal{W}(\tau_k, \tau_{i\sim j}) \tag{1}$$

In our paper, low distance between task denotes high similarity, which guides our task selection.

### 4.2.1 Implementation of Exploration-Exploitation Process in SimMAC

During the Exploration Stage, we deploy multiple RL agents trained uniformly across the task space, systematically collecting trajectory data and measuring convergence points to quantify both task difficulty ($c_\theta$) and occupancy distributions ($\rho^{\pi}_{\mathcal{T}^{\theta}}$). These measurements provide the empirical foundation for our similarity metrics.

In the subsequent Exploitation Stage, we leverage these metrics to construct optimal learning sequences. Drawing inspiration from spiral curriculum (Bruner, 2009), we design a process that systematically builds upon existing knowledge while incrementally increasing difficulty. Beginning with the task exhibiting the lowest convergence point ($\min_\theta c_\theta$), we iteratively select subsequent tasks that maximize similarity to the accumulated experience, formally selecting $\mathcal{T}^{\theta_{j+1}}$ to minimize $d(\mathcal{T}^{\theta_{j+1}}, \mathcal{T}^{\theta_{1\sim j}})$ while ensuring a gradual progression in difficulty. This implementation enables the creation of personalized curricula that maintain coherent knowledge pathways while systematically introducing more challenging concepts, thereby optimizing both learning continuity and skill development.

## 5 Human Subjects Experiment Design

We evaluate our RL-supported teacher algorithms against common baselines (Doroudi et al., 2019) in human learning domains, using human participants who undergo training in the Jumper and Emergency Response games. All studies received local IRB approval. Further details of the environments and the experiment procedure can be found in Appendix.

**Jumper Environment** The Jumper Environment is a 2D obstacle course game developed in Unity (Juliani et al., 2020), inspired by classic platformers. Players navigate a character through spiked pathways using keyboard controls, aiming to reach the level's end without collisions (Figure 14). The environment has two adjustable parameters $\theta$ for level generation: *spike density* and *ground roughness*; these parameters directly influence the difficulty of the level, enabling systematic study of learning progression and adaptive difficulty.

Participants were recruited through an online chat group connecting researchers and screened for device compatibility. To control for prior gaming experience, participants rated their familiarity with 2D side-scrolling games (e.g., Super Mario Bros) to balance experimental conditions.

First, participants received visual instructions on the Jumper gameplay and a trial to familiarize themselves with the controls. After the trial, participants were randomly assigned to one of three conditions:

1. No Training (Control): Participants received no training and proceeded directly to the test stage after the trial. ($n = 80$)

2. Random: Participants played randomly generated training levels. ($n = 78$)

3. PERM-H: Participants received training levels generated by a Jumper-tuned model trained on RL data. The model adapted level difficulty based on inferred player ability. ($n = 72$)

In the Random and PERM-H conditions, participants received 10 different levels with a maximum of 15 attempts per level. Upon completing a level or exhausting attempts, participants progressed to the next level. Finally, after the respective training intervention, they would receive a test level on which we use to measure post-training performance. We initially recruited 240 participants for our study, and filtered out low-effort participants. Finally, there were no significant differences in prior gaming experience across groups (one-way ANOVA: $F(2, 237) = 0.902, p > .05$).

To further investigate the effectiveness of our approach, we conducted a follow-up study comparing PERM-H to a handcrafted curriculum. This handcrafted curriculum, designed by our research team, featured a fixed sequence of training levels with increasing difficulty. We recruited 120 participants via Prolific[1], representing a different sample group from the initial study. After excluding outliers, our final counts were 52 participants in the PERM-H group and 61 in the Handcrafted group. Results from this follow-up study are presented separately from the main study to distinguish between participant pools.

**Emergency Response Environment**   We present a 3D Emergency Response Environment[2] simulating time-critical medical care scenarios (Figure 15). Developed with paramedic services, this environment requires players to select and apply appropriate treatments to patients with evolving conditions during hospital transport. The simulation features stochastic patient state transitions, real-time feedback, and contextual tool information, replicating the decision pressure faced by emergency medical personnel while allowing limited attempts per intervention.

We conducted an experiment with 121 participants, randomly assigned to one of the four groups:

1. Reading Only (control): Learned solely through reading materials, without engaging in gameplay. ($n = 31$)
2. Random: Played tasks selected at random from the pool, without replacement. ($n = 30$)
3. Handcrafted: Followed a predefined task sequence designed by the research team. ($n = 30$)
4. SimMAC: Experienced an adaptively curated task order generated by SimMAC. ($n = 30$)

Except for the Reading group, all participants completed all 17 unique tasks within 45 minutes after a 25-minute reading session on medical knowledge. After the respective treatments, participants were given a multiple-choice questionnaire to assess their knowledge of appropriate measures to take in a medical emergency. One-way ANOVA confirmed no significant differences in prior game experience ($F(3, 117) = 1.34, p = .27$) or emergency handling experience ($F(3, 117) = 1.88, p = .14$) across groups.

## 6 EVALUATION

In our evaluation, we investigate three key research questions: differences in post-training performance across conditions, distinguishing characteristics between curricula, and fundamental differences between RL agents and human learners. For all statistical tests described, we used $\alpha = 0.05$.

### 6.1 POST-TRAINING EVALUATION

We analyzed the effectiveness of teacher-guided training in improving post-training performance on the final test. In Jumper, competence was measured by fewer attempts to complete the test level. In Emergency Response, we counted correct responses on the final multiple-choice test.

**Jumper Environment**   A one-way ANOVA revealed significant differences in final test attempts across groups, $F(2, 237) = 16.461, p < .001$, partial $\eta^2 = .122$, signifying a moderately large effect. Tukey's HSD post-hoc test showed significant differences between No Training and PERM-H ($\Delta\mu = -2.599, p < .001$) and between Random and PERM-H ($\Delta\mu = -1.380, p < .001$). No significant difference was found between the No Training Group and Random Group ($\Delta\mu = -1.219, p = .115$).

---

[1] https://www.prolific.com/

[2] Medical content from West Virginia Department of Health and Human Resources (https://www.wvoems.org/), verified by medical experts during IRB approval.

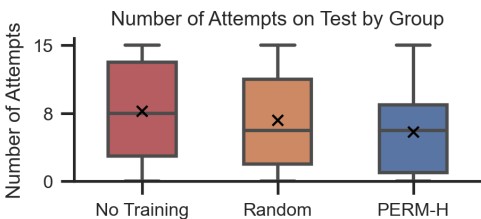 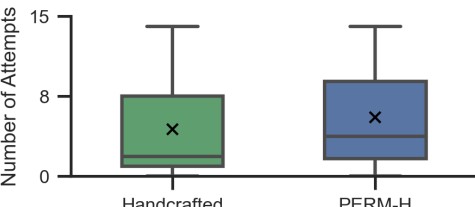

Figure 1: Number of attempts across different conditions for Jumper test. Lower numbers denote better performance. 'X' represents mean number of attempts.

**PERM-H vs. Handcrafted Training** An independent-samples t-test comparing PERM-H ($\mu = 5.904$, $\sigma = 5.558$) and Handcrafted ($\mu = 4.705$, $\sigma = 5.022$) conditions on the Jumper post-training test results showed no significant difference, $t(112) = 1.193, p = .235$, with Cohen's $d = .23$, suggesting a small effect size.

**Emergency Response Game** A one-way ANOVA showed significant differences in the test scores among groups, $F(3, 117) = 12.46$, $p < .001$, partial $\eta^2 = .24$, signifying a large effect. Tukey's HSD post-hoc comparisons revealed significant differences between SimMAC and both random ($\Delta\mu = -3.21$, $p < .001$) and reading-only conditions ($\Delta\mu = -3.53$, $p < .001$). The handcrafted condition also differed significantly from random ($\Delta\mu = -1.81$, $p = .03$) and reading conditions ($\Delta\mu = -2.13$, $p = .009$). No significant differences were found between SimMAC and handcrafted conditions ($\Delta\mu = -1.40$, $p = .155$) or between random and reading conditions ($\Delta\mu = -0.326$, $p = .960$).

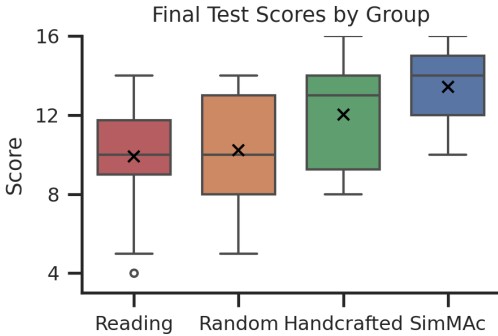

Figure 2: Results of Emergency Response knowledge test. 'X' denotes mean score on test.

In summary:

1. Students trained using our proposed teacher algorithms significantly outperformed those in the control and Random curricula groups in both environments.

2. Students trained under the handcrafted curriculum also outperformed those in the control and Random curricula groups.

3. No significant performance difference was observed between students trained with our algorithms and those trained with the Handcrafted curriculum. Similarly, no significant difference was found between the Random and control groups.

The results for Jumper and Emergency Response game are visualized in Figure 1 and 2 respectively.

**Discussion** These findings demonstrate that our RL-bootstrapped teacher algorithms (PERM-H and SimMAC) significantly outperformed both random and control curricula groups while achieving comparable results to expert-designed curricula—despite requiring no manual design effort. Overall, these results lend credibility to the efficacy of algorithms supported by RL agents in curriculum design. Surprisingly, the Random group showed no improvement over the No Training group despite greater domain exposure, highlighting that unstructured practice offers minimal benefit and reinforcing the value of intelligently sequenced learning experiences.

## 6.2 COMPARISONS TO EDUCATION BASELINES

Given the central focus on level difficulty (PERM-H) and task similarity (Sim-MAC) in the respective environments, we draw comparisons between our proposed teacher algorithms and baselines in the context of these metrics.

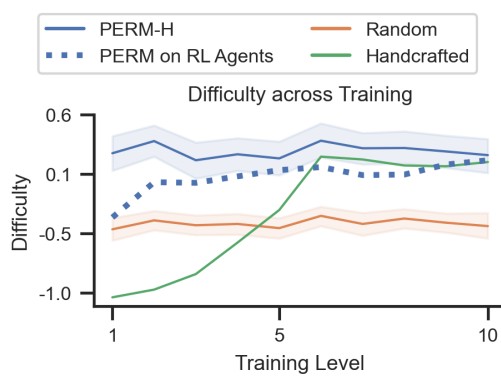

Figure 3: Difficulty progression across curricula for Jumper. PERM-H introduces challenges earlier than alternatives. RL agents reach difficulty levels comparable to humans, supporting their viability as warm-start learners.

**Jumper** Figure 3 shows PERM-H-generated levels consistently exhibited higher difficulty compared to random curricula. This rigorous training benefited students when encountering the complex final test level. Contrary to expectations of a logarithmic training curve with initial growth followed by plateauing, such as the one exhibited by the Handcrafted group, PERM-H participants faced challenging environments early, resulting in a performance ceiling effect. Many PERM-H group participants appeared to reach this upper bound during training due to the Jumper domain's relative simplicity. PERM-H demonstrated the ability to quickly infer learner ability levels and present challenging levels early in training, contrasting with the random curriculum's potentially wasted training opportunities.

The Handcrafted curriculum began with extremely easy levels, slowly increasing difficulty to reach a plateau comparable to PERM-H's level around the 5th training level. Compared to the adaptive curriculum provided by PERM-H, this suggests that initial levels provided minimal training value, and participants could have benefited from a shorter, more efficient training regimen beginning at a higher difficulty level.

**Emergency Response** Figure 4 illustrates the cumulative distance during training under SimMAC-generated and Handcrafted curricula, calculated by Equation 1. The SimMAC curriculum results in a lower cumulative distance throughout training compared to both Random and Handcrafted curricula. The Random curriculum's cumulative distance is similar to the Handcrafted curriculum but less effective due to higher variation in task similarity and lack of easy-to-hard ordering. Students' better performance under the SimMAC curriculum indicates that emphasizing learning continuity and smoother experiences leads to positive learning outcomes.

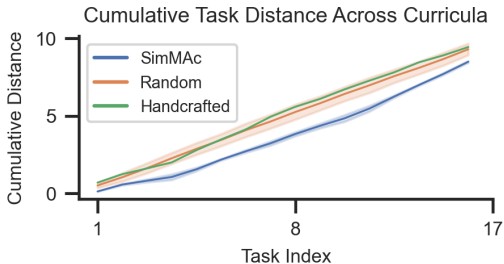

Figure 4: Cumulative distance comparisons across different curricula for Emergency Response. Higher distance means lower similarity.

## 6.3 COMPARISONS TO RL AGENTS

This section attempts to investigate whether RL agents are suitable as warm-start human learners by comparing RL Agent and human training.

**Jumper** We trained a PPO (Schulman et al., 2017) student agent using PERM as the teacher algorithm for 24,000 episodes. Figure 3 compresses the 24,000 RL training episodes into 10 levels, matching the human training scale. As training progresses, the artificial student agent encounters increasingly challenging environments, ultimately reaching difficulty levels comparable to handcrafted levels and, to some extent, humans trained under PERM-H.

**Emergency Response** For each task-pair $i, j$, we calculate the Wasserstein distance between performance distributions for both RL agents and human students, and plotted these paired distances in Figure 5, right. A Pearson correlation coefficient was computed to assess the relationship between them, and we found a moderate positive correlation between the two variables ($r = .490, n = 287, p < .001$).

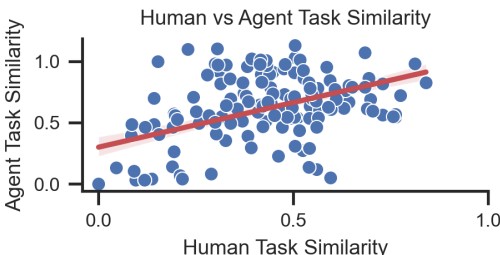

Figure 5: Inter-task similarity as derived from agents versus humans. Red line represents a regression line of $r = .490$

**Discussion** Our findings across two environments demonstrate both the potential and limitations of using RL agents as warm-start human learners. In the Jumper environment, we corroborate the results of Tsividis et al. (2017), with humans demonstrated superior learning efficiency, reaching high performance levels quickly while RL agents required millions of experiences to achieve even minimal human performance levels. Despite this gap, RL agents and humans showed consistent agreement on task difficulty rankings. The alignment suggests that in carefully designed domains, RL can effectively provide valid initial training data in place of human learners.

In the Emergency Response domain, a moderately positive correlation emerged between inter-task similarities derived from humans and agents, indicating some alignment between artificial and human learning patterns. Notably, when selecting tasks during human trials, we relied on the distance between human task trajectories and task trajectories, without updating the similarity metrics with human data. Despite this direct comparison of task similarity from artificial to human learners, the approach yielded excellent learning outcomes, demonstrating RL agents' effectiveness as warm-start substitutes for human learning data.

While differences between human and RL agents persist across both domains, our findings highlight both the current limitations of RL in matching human learning efficiency and its potential to inform and enhance human learning processes. The ability to automatically collect training data without expert intervention, combined with positive student outcomes, justifies our approach of using RL agents to train teacher algorithms. This lays the groundwork for developing more sophisticated adaptive learning systems.

## 7 CONCLUSION AND FUTURE WORK

We investigated using RL agents as warm-start proxies to address the cold-start problem in teacher algorithms. Our approach trains PERM-H and SimMAC through structured Exploration and Exploitation stages. Human studies showed that our RL-bootstrapped curricula outperformed baseline methods and matched expert-designed curricula without requiring extensive human data or domain expertise.

While our findings suggest a viable pathway for reducing initial data dependencies in adaptive learning systems, our approach is not without limitations. First, our approach is currently constrained to environments that can effectively model both RL and human learning patterns, and notable alignment gaps exist between these modalities. Second, our analysis revealed that RL agents has distinct differences from human learners, suggesting the need for better alignment techniques.

Future work should investigate methods to better calibrate and evaluate the gap between RL agent behavior and human learning patterns, perhaps through transfer learning approaches or hybrid models that incorporate limited human data earlier in the process. Additionally, researchers might explore how this bootstrapping methodology generalizes across more diverse learning domains, particularly those with abstract reasoning requirements or social components. We invite the community to build upon our testbed environments to develop improved alignment metrics and evaluation frameworks, potentially expanding this approach to broader educational contexts. As this nascent field develops, integrating generative AI with RL-based curriculum design could open new avenues for creating more accessible, effective, and personalized learning experiences.

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

# A APPENDIX

## A.1 FURTHER DETAILS ON TEACHER ALGORITHMS

### A.1.1 PERM-H

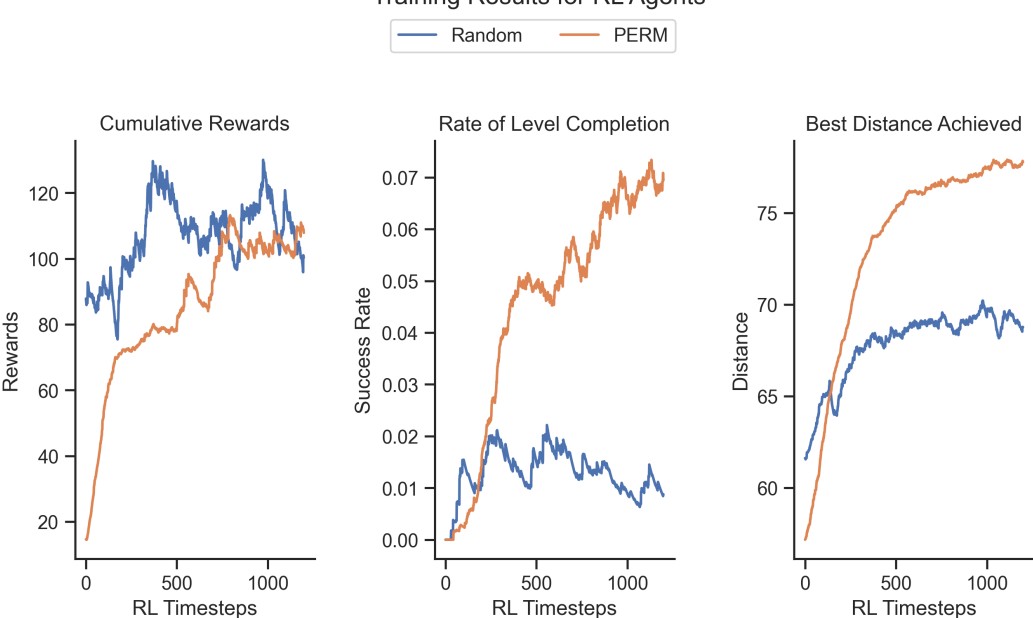

Figure 6: Training results of RL Agents trained under PERM (orange) and a random curricula (blue). Left: Agents trained under PERM-H increased in ability over time, despite levels of increasing difficulty. Centre: PERM trainees are more likely to complete the level than those under random. Right: Agents trained under PERM travelled deeper into the level than the counterparts in the random condition.

**Pre-study** To determine if PERM applies well to our Jumper environment, we conducted a pre-study in which we use PERM to train a student RL agent.

We first train a Jumper-tuned version of PERM. For the Jumper environment, we collected a tuple of (*spike density, height variance, rewards*) for every episode of the RL training. In this development phase, we obtained a total of 14506 environment-student interaction data, over a course of 12 hours, with a single V100 GPU. Thereafter, we deploy the trained PERM-H as a teacher algorithm to a new PPO Schulman et al. (2017) RL student trained using Unity's `ml-agents` package Juliani et al. (2020). We also provide the results of a RL student trained under a random curricula. The results are shown in Figure 6.

Based on the obtained results, it is evident that the adoption of an Item Response Theory-driven curriculum with the PERM teacher yields remarkable outcomes for RL agents, surpassing the performance achieved by the random curriculum. Notably, RL agents trained using the IRT-driven curriculum exhibit a higher level of proficiency in completing levels and, on average, traversed deeper into these levels compared to their counterparts trained using the random curriculum. These impressive outcomes are noteworthy considering that PERM continually challenges the student by evolving the levels in the same pace.

**Futher Analysis on Performance** We compared participant's completion rate. We also compared participant's self-reported familiarity with side-scrolling games against their completion rates. A successful completion meant that participants took lesser than 15 attempts on the final test. Lastly, we analyzed the duration it took per attempt for them to complete. We perform the above analysis based on the assumption that more competent participants would complete the test with lesser at-

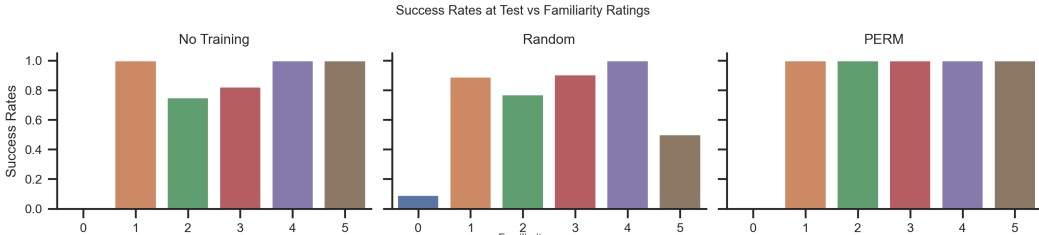

Figure 7: Participant's self-report of their familiarity with 2D games, against their completion rates in the final test. A score of 0 represents "No Experience at all" while 5 represents "Highly Experienced". All participants under PERM-H were successful in completing the test, with the exception of individuals who had "No experience at all" in 2D Games.

| Name | Jumper |
|---|---|
| **Environment Type** | UED |
| **Short Description** | A Super-Mario inspired 2D game, where players have to control a character to jump across obstacles to reach the end |
| **Student Objective** | Reach the end of the level, while avoiding obstacles |
| **Student Actions** | Keyboard controls to control main character's movement and jumping |
| **Env Parameters to adjust** $\theta$ | Spike Density; Ground Roughness |
| **Skills Imparted** | Motor-skills, hand-eye coordination |

Table 1: Overview of Jumper Game Environment

tempts, with a shorter duration. We used Student's t-test to compare the duration and the attempts made in the final test, and chi-squared test of goodness of fit to compare completion rates.

**Results** The completion rate of the tests are presented in Figure 7. Participants under the PERM-H were more likely to complete the test (i.e. reach the goal with less than 15 attempts), regardless of prior experience with games, than the other conditions. Figure 7 depicts the completion rate of each condition, compared to their self-reported prior experience. The effect of curriculum was found to be significant, i.e. the completion rates were not equally distributed amongst the 3 conditions ($\chi^2(2, N = 230) = 9.24, p < 0.01$).

Lastly, the duration per attempt for groups under PERM-H ($\mu = 61.02, \sigma = 66.41$) were significantly

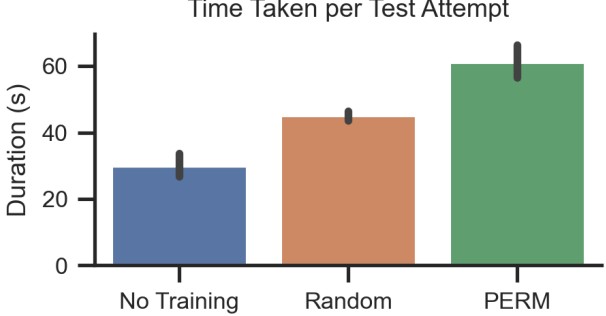

Figure 8: Participants under PERM-H took a longer time per attempt during the test ($p < 0.01$).

longer than that of the random curricula ($\mu = 45.01, \sigma = 19.68, p < 0.01$) and control condition ($\mu = 29.86, \sigma = 16.42, p < 0.01$). The average duration is plotted in Figure 8.

**Discussion** Collectively, these findings suggest that students trained with PERM-H were not only more likely to succeed on the test but also required fewer attempts to do so. Crucially, this positive impact of PERM-H on students remains consistent across individuals with diverse levels of prior experience with similar games. This consistency underscores the effectiveness of the adaptive curriculum implemented by PERM-H, demonstrating its capacity to benefit participants regardless of their varied backgrounds.

| Name | Emergency Response |
|---|---|
| **Environment Type** | Task Sequencing |
| **Short Description** | A Overcooked-inspired game, where players take the role of a paramedic providing medical assistance to a patient enroute to the hospital |
| **Student Objective** | Provide the necessary medical assistance, in reaction to a description of patient's conditions |
| **Student Actions** | Mouse to control paramedic's movement, and to guide and pick up the necessary medical devices |
| **Env Parameters to adjust $\theta$** | Task from a pre-determined pool |
| **Skills Imparted** | Medical knowledge and decision making, working under time pressures |

Table 2: Overview of Emergency Response Game Environment

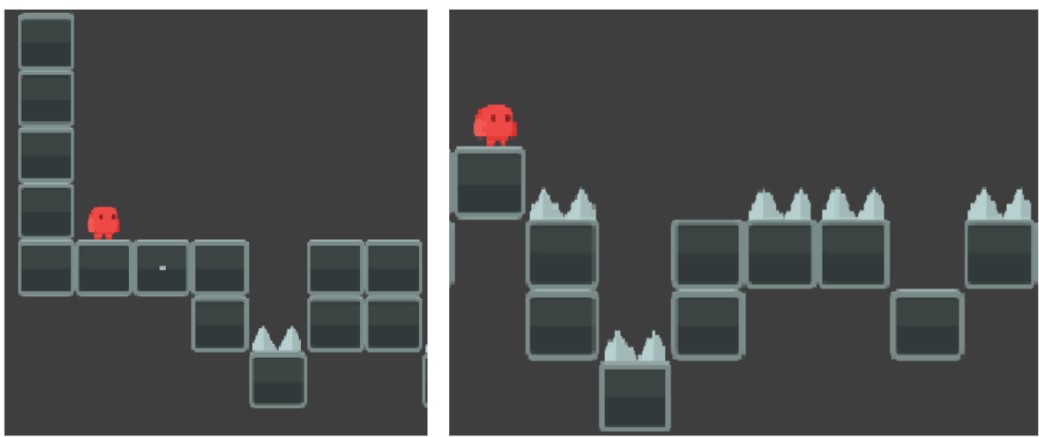

Figure 9: Possible segments of levels generated by PERM-H. The easy level (left) has lesser spikes and lesser variation in the terrain. In contrast, players have to navigate uneven terrains and jump across more spikes in the difficult level (right).

We were surprised that students under PERM-H had took significantly longer per attempt to complete the test. This observation hints at distinct behavioral differences among the learners, especially those exposed to higher difficulty levels. It's worth highlighting that participants were not explicitly informed that their performance was being evaluated based on the speed of level completion. This absence of explicit information could have influenced the more deliberate approach adopted by students exposed to the PERM-H framework.

**Enjoyment During Training**

**Method** At the end of the training trial, we conducted a short survey that queried participants on how fun they found the training.

**Results** Participants assigned to the PERM-H condition rated the game as less fun ($\mu = 3.18, \sigma = 1.06$) as compared to participants in the no training condition ($\mu = 3.43, \sigma = 1.16, p = 0.027$) but not significantly different from the participants in the random curricula ($\mu = 3.29, \sigma = 1.29, p = 0.044$).

**Discussion** We noticed that participants who did not undergo any form of training tended to rate the game as more enjoyable than those who received training. This disparity in enjoyment levels might be linked to the potential fatigue induced by the training process. A closer analysis showed that, on average, both participants with average ($\mu = 4.08, \sigma = 2.98$) performance under the PERM-H framework required more attempts to complete their training compared to their peers in the ran-

dom curricula ($\mu = 3.43, \sigma = 2.28, p < 0.01$). It's important to note that this increased number of training attempts was a desired outcome of PERM-H, as it consistently provided levels within the grasp of the participant's ability.

## A.2 SimMAC

In this section, we provide more details of the SimMAC algorithm and related backgrounds of SimMAC.

**Background: Wasserstein Distance**   Wasserstein distance was employed to estimate the distance between two tasks in DIPLR Li et al. (2023a). DIPLR focuses on the pair-wise distance and calculates the distance between two tasks $d(\mathcal{T}^{\theta_1}, \mathcal{T}^{\theta_2})$ as:

$$
\mathcal{W}(\rho_{\mathcal{T}^{\theta_1}}^{\pi}, \rho_{\mathcal{T}^{\theta_2}}^{\pi}) = \left( \inf_{\psi \in \Pi(\rho_{\mathcal{T}^{\theta_1}}^{\pi}, \rho_{\mathcal{T}^{\theta_2}}^{\pi})} \mathbb{E}_{(\phi_1, \phi_2) \sim \psi}[d(\phi_1, \phi_2)^p] \right)^{1/p}
\tag{2}
$$

where $\phi \in (S, A)$ is a sample from the occupancy distribution. By Equation (2), DIPLR collects state-action samples in trajectories to compute the empirical Wasserstein distance between two tasks. I.e., $d(\mathcal{T}^{\theta_i}, \mathcal{T}^{\theta_j}) \mathcal{W}(\rho_{\mathcal{T}^{\theta_i}}^{\pi}, \rho_{\mathcal{T}^{\theta_j}}^{\pi}) \approx \mathcal{W}(\tau_i, \tau_j)$ is our empirical estimation of the Wasserstein distance between two tasks.

We extend the methodology in DIPLR and employ Wasserstein distance to calculate the distance between one task and a set of tasks, $d(\mathcal{T}^{\theta_k}, \mathcal{T}^{\theta_{i \sim j}})$:

$$
\mathcal{W}(\rho_{\mathcal{T}^{\theta_k}}^{\pi}, \rho_{\mathcal{T}^{\theta_{i \sim j}}}^{\pi}) = \left( \inf_{\psi \in \Pi(\rho_{\mathcal{T}^{\theta_k}}^{\pi}, \rho_{\mathcal{T}^{\theta_{i \sim j}}}^{\pi})} \mathbb{E}_{(\phi_1, \phi_2) \sim \psi}[d(\phi_1, \phi_2)^p] \right)^{1/p}
\tag{3}
$$

**Exploration Stage**   During the Exploration Stage of SimMAC, we initialize a diverse set of RL agents and train them uniformly on all tasks. We collect the trajectories at different stages during training such that the agent trajectories have a wide coverage over each task and we can use them to obtain a good occupancy measure for each task. Assume we have $k$ tasks and we denote the trajectories associated with each task by $\Gamma^1, \Gamma^2, ..., \Gamma^k$. The complete procedures of the SimMAC algorithm are summarized in Algorithm A.2.

[th] SimMAC for Emergency Response Game $k$ training tasks: $\mathcal{T}^{\theta_1}, \mathcal{T}^{\theta_2}, ..., \mathcal{T}^{\theta_k}$, training curriculum length $N$ ($N \leq k$), empty trajectory buffer $\Gamma$

Measure the difficulty of each task
Select task with the lowest difficulty, denoted by $\mathcal{T}^{\theta_1}$
Train human learner in $\mathcal{T}^{\theta_1}$ and collect the trajectories, $\tau_1 \sim \mathcal{T}^{\theta_1}$
Insert $\tau_1$ into $\Gamma$
$t = 2, 3, ..., N$ $i = 1, 2, ..., N$ Calculate task similarity between $\mathcal{T}^{\theta_i}$ and the rest of the tasks by $d = \mathcal{W}(\Gamma, \Gamma^i)$
Select the task with the lowest distance, denoted by $\mathcal{T}^{\theta_t}$
Train the human learner in $\mathcal{T}^{\theta_t}$ and collect the trajectories, $\tau_t \sim \mathcal{T}^{\theta_t}$
Insert $\tau_t$ into $\Gamma$

**Qualtitative Feedback from Participants**   At the end of the experiment, we conducted a short survey to gather participants' feedback on how enjoyable they found the game, the coherence of their learning experiences, and whether they felt fatigued afterward. Our primary focus was on their feedback regarding the consistency and coherence of the curriculum.

Participants in the Random group frequently complained about the lack of coherence in their learning experience, as tasks were randomly shuffled, leading to a disjointed progression for some. In contrast, participants in the SimMAC group reported a more coherent and continuous learning experience.

In addition to smooth knowledge accumulation, human learners showed a strong preference for progressing from easy to more difficult tasks. This preference is interesting because it contrasts with what is typically effective for training reinforcement learning (RL) agents. In RL, numerous studies Wang et al. (2019); Dennis et al. (2020); Jiang et al. (2021); Parker-Holder et al. (2022) highlight the benefits of training in novel and challenging environments. This difference in learning preferences can be attributed to the distinct objectives and constraints in RL training versus human training. In RL, the goal is to develop agents with general capabilities that can transfer to unseen challenges, often involving billions of training timesteps. On the other hand, human training emphasizes maximizing learning efficiency within a limited timeframe, as extended curricula can lead to fatigue.

### A.2.1 EXTENDED EXPERIMENT RESULTS

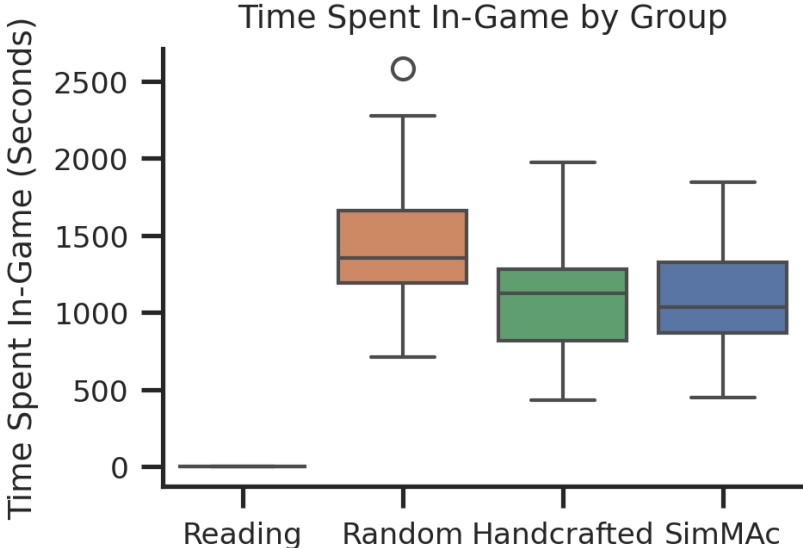

Figure 10: Game time by various groups.

All participants were compensated for their participation in our study, at a rate that is above or the same as Prolific's recommended payment principles (https://researcher-help.prolific.com/en/article/2273bd).

**Game Time** Figure 10 compares the game time across three different experimental groups: Handcrafted, SimMAC, and Random. The Reading group is the control group, which did not participate in the game but instead focused on reading materials related to emergency response knowledge. Key observations include:

1. The SimMAC group, which used the proposed SimMAC teacher for curriculum training, has a median game time of about 18 minutes, with a relatively tight interquartile range (IQR) from around 15 to 22 minutes. This suggests that participants in this group were able to complete the game efficiently.

2. The Handcrafted group shows a similar median game time, also around 18 minutes, but with a slightly wider IQR compared to the SimMAC group. This indicates a bit more variability in performance.

3. The Random group has the highest median game time, approximately 22 minutes, with the broadest IQR, suggesting greater variability in how long participants took to complete the

game. There is also an outlier, indicating that at least one participant took significantly longer than others.

In summary, the results highlight the effectiveness of the SimMAC teacher in providing a training curriculum that allows human learners to complete the task more efficiently, as evidenced by the lower game times. Moreover, participants in the SimMAC group achieved the highest post-test scores, demonstrating that the efficiency gained in game time did not come at the cost of learning quality.

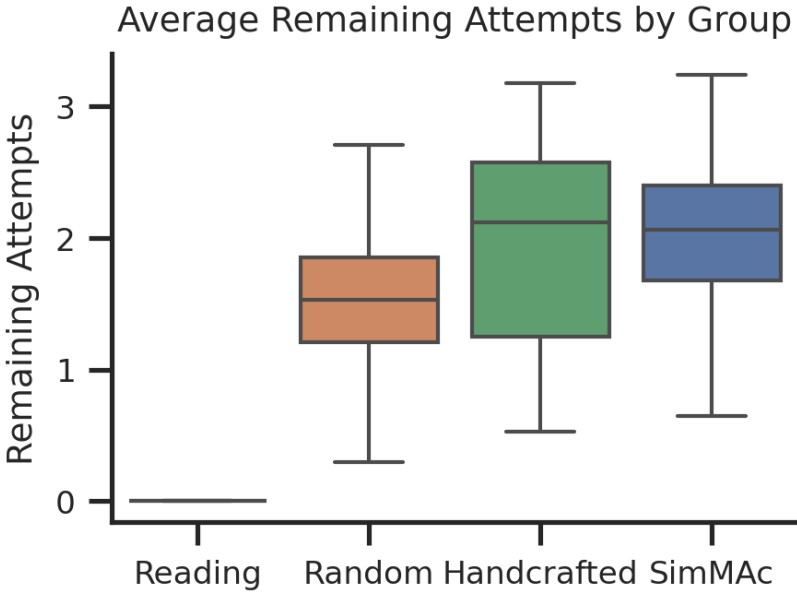

Figure 11: Averaged remaining attempts in each task during the game.

**Remaining Attempts in the Game** Figure 11 provides the average remaining attempts in each task during the game. In general, participants in Random group required more attempts to complete the scenario. SimMAC and Handcrafted, on the other hand required lesser attempts. This can be attributed to the easy-hard progression that is a feature of SimMAC and Handcrafted curriculum, so that participants do not face a difficult task even before they have learned about it.

**Participant's Assessment of Fun and Usefulness** After the experiment ended, participants were tasked to complete a survey on their training experience. The results pertaining to the fun factor ("How do you rate the fun factor of the game?") and usefulness of their curricula ("Did you feel the order in which these scenarios were presented to you to play, helped you to learn these scenarios better?") are presented in Figure 12 and Figure 13 respectively. Overall, all participants found the Emergency Response Game fun with average scores well above 3 points ($\mu = 3.78$). Notably, participants were more likely to find the curriculum generated by SimMAC to be helpful.

### A.3 ENVIRONMENT DETAILS

#### A.3.1 EMERGENCY RESPONSE ENVIRONMENT

Our research team designed the emergency response game for paramedic training for non-expert human learners. The participants engaged in our experiment will learn emergency response knowledge through interactive video games.

A clear illustration of the game interface is presented in Figure 15. In the game, the human player navigates the ambulance, selecting appropriate medical items to treat patients with various conditions. The patient's condition transitions stochastically, meaning it can change to different states

Figure 12: Averaged remaining attempts in each task during the game.

Figure 13: Averaged remaining attempts in each task during the game.

after the application of a particular medical item. The current condition of the patient is displayed in the top right corner, and this description updates dynamically as the condition evolves. When the mouse hovers over a specific medical item, a description of the item and its functions appears in the bottom right corner.

Players must complete a series of treatments to stabilize the patient before the ambulance reaches the hospital. Our research team designed 10 different medical conditions, including *Allergy*, *Seizure*, *BreathingDifficulty*, *HeatStroke*, *ExternalBleeding*, *ColdExposure*, *AbdominalTrauma*, *Musculoskeletal­Trauma*, *AcuteCoronarySyndrome*, *Bronchospasm*. Two of these conditions (*Seizure* and *ColdExposure*) were used to create a demo video to instruct participants on gameplay. The remaining conditions form the task pool for training. Depending on the natural complexity of each condition, we developed easy, medium, and hard versions for some diseases. However, conditions

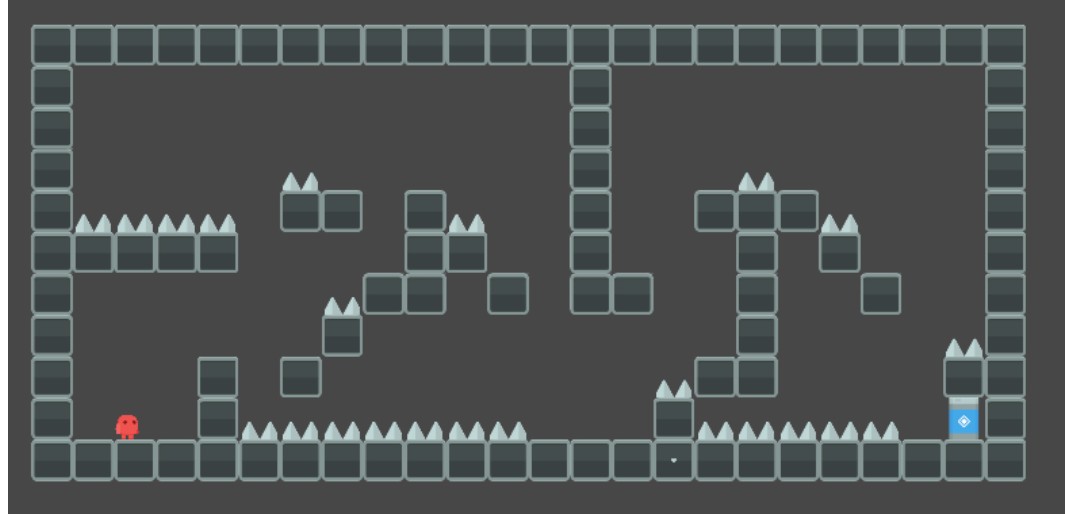

Figure 14: Jumper Game's test level. Players control the red figure to navigate the spiked maze, with the objective of reaching the final goal in blue.

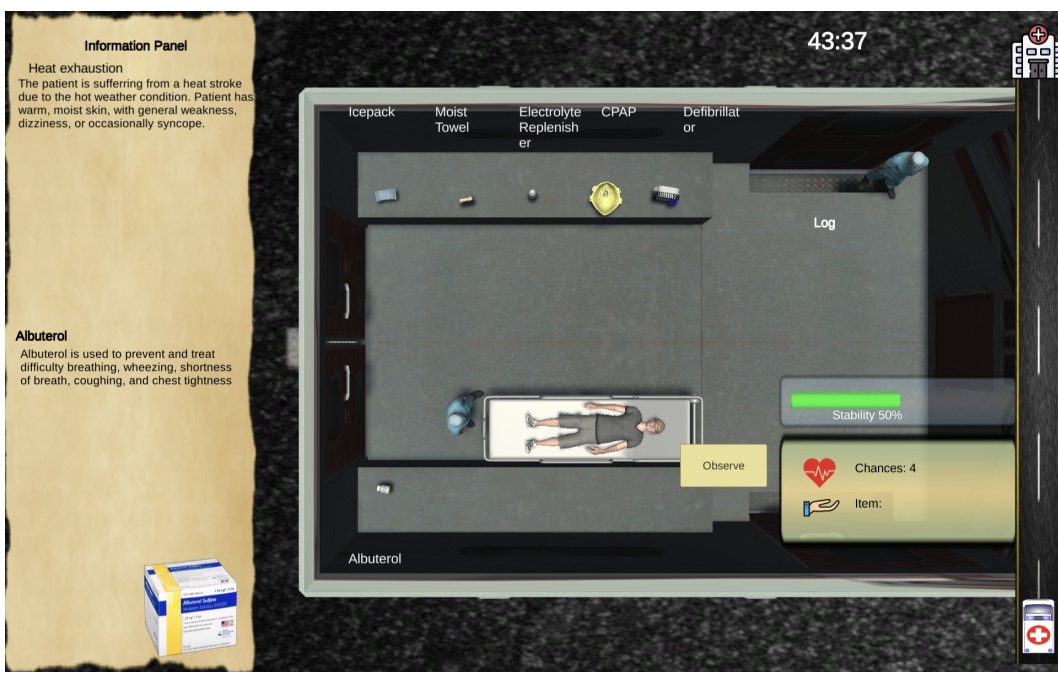

Figure 15: Blown-up version of the Emergency Response Game, providing a bird-eye view of the interior of an ambulance enroute to the hospital. Participants have to control the medical officer (in blue) to retrieve appropriate medical equipment to address patient's condition. The Information Panel on the left describes the patient's condition, and a short description of the item when participant's mouse hovers over an item.

like *ExternalBleeding* and *HeatStroke* may have only easy or medium versions due to a lack of diverse condition variations. In total, 17 tasks were constructed to form the training curriculum.

Figure 16 presents a segment of the flowchart for the *BreathingDifficulty* condition. For instance, in the stochastic transition, the patient's state can evolve to either *patient-state=1* or *patient-state=10* after the player applies CPAP. The player navigates the flowchart by selecting different actions (i.e., medical items) and eventually reaches various termination states. Condition variations refer to differ-

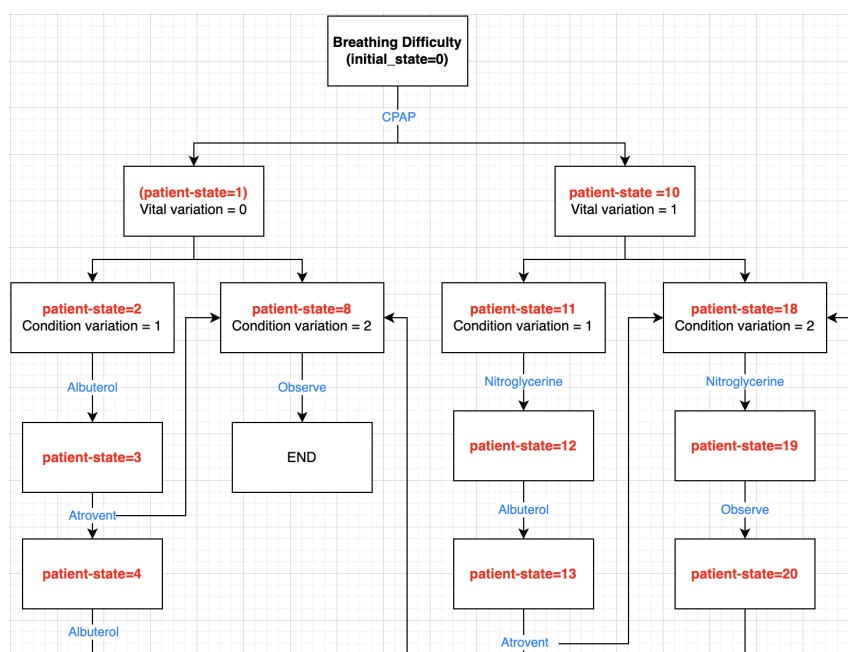

Figure 16: Flowchart of the *BreathingDifficulty* disease.

ent severities of the same disease, such as mild *HeatStroke* versus severe *HeatStroke*. Vital variations involve changes in vital signs, like blood pressure and body temperature, which influence the treatment approach. Additionally, vital variations trigger dynamic updates in the game, displaying the relevant vital value and range (indicated by the green bar). Through this interactive game, players progressively accumulate knowledge and skills for handling various emergency response situations.

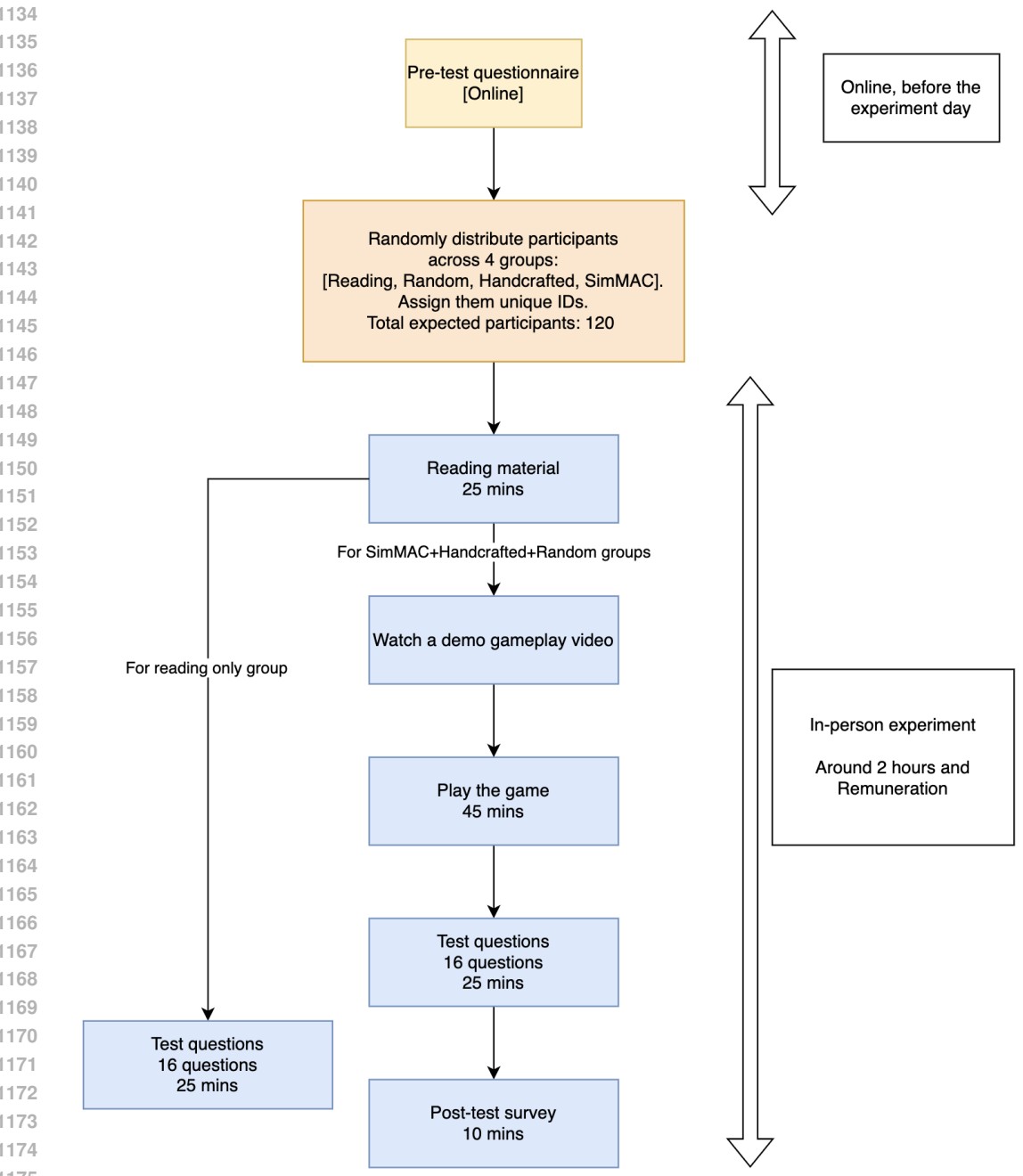

Figure 17: Public Experiment Flow.

### A.3.2 ADDITIONAL PROCEDURES FOR HUMAN SUBJECTS TRAINING

Based on feedback from 8 volunteer testers, we adjusted our experimental setup. We reduced the number of diseases from 10 to 8 and decreased total tasks from 21 to 17 to mitigate participant fatigue. We also added 2 simpler tasks for a demo video and warm-up to familiarize participants with the game. Figure 17 illustrates the detailed experiment flow.

Pilot test feedback revealed participants prefer completing one topic before moving to another, even if tasks in new topics have higher similarity to past experiences. Consequently, we adjusted Sim-MAC to complete all tasks within a current condition before introducing a new one.

The participants' initial reading materials were adapted from West Virginia Department of Health and Human Resources[3]. Prior to the commencement of the study, the research team had consulted a medical expert and they had confirmed that the medical information provided above are not misrepresented, even in the local context, and poses no harm to the participants. As an added measure, participants were debriefed after the experiment and explicitly advised to disregard the session as indicative of local medical emergency protocols. They were directed to context-specific online resources for more localized information.

### A.4    PARTICIPANT BACKGROUND ANALYSIS

#### A.4.1    EMERGENCY RESPONSE GAME

We conducted a comprehensive ablation analysis to ensure that the performance of the SimMAC curriculum is not influenced by participants' backgrounds. Most participants in our experiment were university students with similar demographics, including age, learning abilities, reading skills and etc. We focused on three key factors: whether participants held a job related to healthcare, their experience with 3D games, and their initial proficiency in emergency procedures.

**Healthcare Job**    Participants with healthcare-related jobs might perform better during the game and in post-test questionnaires. Therefore, we collected this background information in the pre-test questionnaire and summarized the job backgrounds of all participants in Figure18.

**3D Game Experience**    Experience with 3D games could also influence performance. The distribution of 3D game experience by group is shown in Figure 19.

A two-way ANCOVA was conducted to examine the effects of Group assignment and Game Experience on the final test scores, with Game Experience serving as a covariate. The analysis revealed a significant main effect of Group $(F(3, 113) = 10.32, p < .001)$. However, the covariate, Game Experience, did not show a significant effect $(F(1, 113) = 1.79, p = .183)$. The interaction between Group and Game Experience was also not statistically significant $(F(3, 113) = 0.07, p = .974)$.

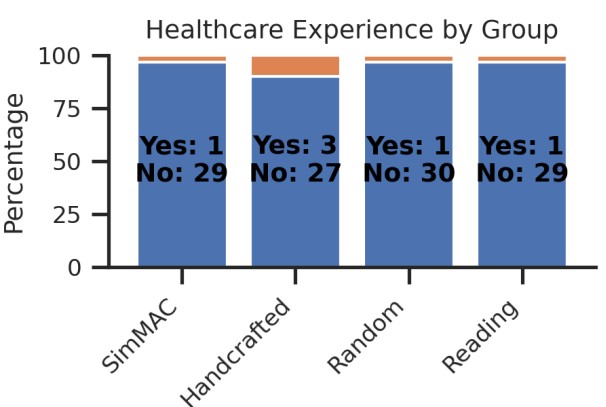

Figure 18: Participants' background of healthcare job.

In summary, our experiment design was successful in mitigating for prior experience in games as a potential confounding factor for our final test scores, and thus was not discussed in the main text.

**Proficiency in Emergency Procedures**    Finally, we analyzed participants' proficiency in emergency procedures, i.e., prior knowledge of handling emergency situations, as shown in Figure 20. A two-way ANCOVA was conducted to examine the effects of Group assignment and Emergency Proficiency on test scores, while controlling for Emergency Proficiency as a covariate. The results revealed a significant main effect of Group $(F(3, 113) = 10.34, p < .001)$. There was also a significant effect of the covariate, Emergency Proficiency $(F(1, 113) = 8.92, p = .003)$. However, the interaction between Group and Emergency Proficiency was not statistically significant $(F(3, 113) = 1.49, p = .221)$.

Taken together, it would suggest that while Emergency Proficiency and Group independently influenced the final test scores, Emergency Proficiency was not a confound of group assignment. Our

---

[3]https://www.wvoems.org/

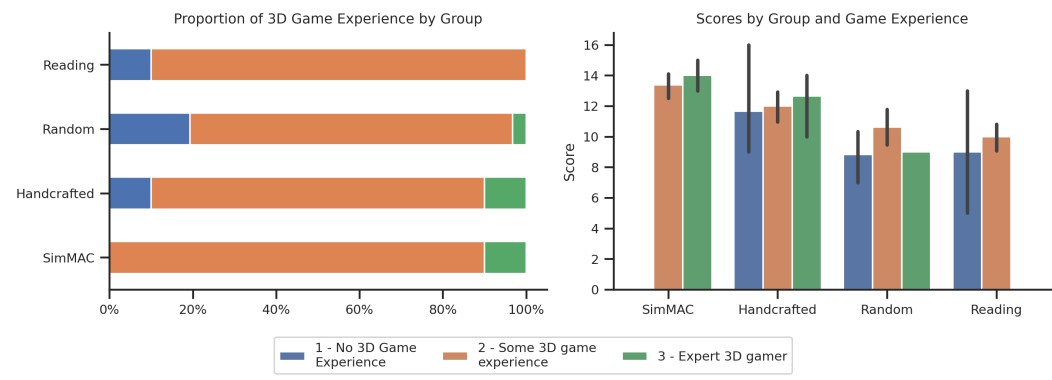

Figure 19: Left: Proportion of self-reported experience with games by Group. Right: Scores by Group and prior Game Experience

experimental procedure had sufficiently controlled for prior experience in Emergency situations and thus was not discussed in the main text.

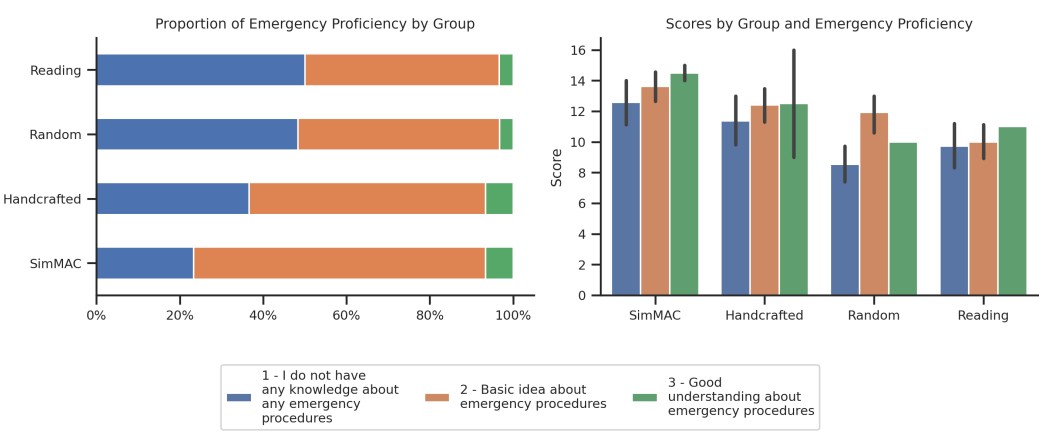

Figure 20: Left: Proportion of self-reported experience with emergencies and medical procedures by Group. Right: Scores by Group and prior experience with medical emergencies.

