# OpenReview forum: "From Machine to Human Learning: Towards Warm-Starting Teacher Algorithms with Reinforcement Learning Agents"
_ICLR.cc/2026/Conference — ICLR 2026 Conference Withdrawn Submission_

### Official Review · Reviewer_FN2d · 2025-10-28

**Soundness:** 4
**Presentation:** 2
**Contribution:** 3
**Rating:** 2
**Confidence:** 4

**Summary:**

The papers studies the of student-teacher teaching problem where an AI teacher has to train humans students to performs tasks though a learning curriculum. To mitigate the cost of expensive human data collection, the authors propose having RL agents act as proxy humans to collect data for the teacher model. The authors introduces two distinct methods that utilizes this proposed framework, PERM-H and SimMAC. The authors then conduct user studies with human subjects to evaluate their proposed methods on two district teaching environments, a 2D platforming game and a top-down game simulating emergency medical response.

**Strengths:**

- The paper tackles the very relevant research problem of Human-AI Teaching.
- The human subjects experiments are very extensive and well-structured.
- The authors fairly address the limitations to their work, highlighting that despite the positive results, there are differences between RL and human learning.

**Weaknesses:**

- While the experiments are extensive, the technical contribution of this paper are unfortunately fairly marginal.
   - There does not seem to be any major difference between PERM and PERM-H other than adding the difficulty to ability scaling, $\epsilon$.
   - For SimMAC the main contribution of using Wasserstein distance to measure task similarity is also adapted directly from an previous work (DIPLR) with minimal changes.
- Some key questions/steps about the proposed methods remain unclear (See Questions)
- As one of the proposed methods, PERM-T is directly adapted from PERM, it would benefit reader is the authors could provide a short overview of PERM in the paper as part of the preliminaries.
- "RL agents and humans showed consistent agreement on task difficulty rankings." (Line 449)
I find this line to be misleading as at no point in the paper does the authors indicated that they asked the humans to rank the difficulty of the generated levels/tasks, they merely shown that a teacher bootstrapped with RL data can effectively teach humans.
- There seems to be a formatting error for Algorithm A.2 and the algorithm itself seems to be incomplete.
- (Minor) The captions for Figures 11, 12 and 13 are repeated.

**Questions:**

- Are there any significant difference between PERM and PERM-H other than the addition of the scaling parameter? Is PERM-T simply trained PERM model but evaluated on human students?
- For SimMAC, how are the RL agents trained? How many agents are trained per task?
- Why is task similarity, rather than difficulty used in Emergency Response/Task Sequencing in selecting the next learning task? This is not entirely clear to me.
- For Emergency Response
   - How do the task differ from each other from and RL training perspective? Do they have district rewards for each task?
   - Are there any intermediary rewards or just rewards for success/failure?
   - How is the complexity of the task determined?
- Do the authors consider the Emergency Response as part of their contribution and if so, do they they plan on open-sourcing the environment?
- Can the proposed approach of using RL agents as bootstrapping applied to any UED method?

---

### Official Review · Reviewer_oGSx · 2025-10-28

**Soundness:** 3
**Presentation:** 3
**Contribution:** 2
**Rating:** 4
**Confidence:** 2

**Summary:**

The paper trying to solve the problem of cold-start in applying AI methods to teaching humans, especially in curricula design. This is sometimes called the "sequencing" problem where the AI methods are used to pick the best sequence of tasks / content for the human learners. The main idea here is to use RL algorithms to generate a dataset and use this dataset as a starting point to have warm-start. The way the paper proposes to do this is to use the generated dataset of tasks and RL algorithms to generate estimates of task difficulty (represented by RL algorithm's convergence time) and task similarity (Wasserstein between state-action occupancy measures). The paper then proposes two approaches to utilize this: one that chooses next task based on difficulty and similarity, and an extension of PERM that relaxes the difficulty = ability assumption. The empirical results test the two proposed approaches against Random and Handcrafted curricula with humans. The proposed methods perform equally good as the Handcrafted curricula (the paper reports no statistically significant difference was found between the two).

**Strengths:**

- The paper addresses an important and timely problem: curriculum design for human learners using reinforcement learning (RL).
- Tackling the cold-start issue in sequencing tasks for humans is a meaningful contribution area.
- The idea of leveraging RL agents to generate a dataset for warm-starting, followed by fine-tuning with humans, is conceptually appealing.
- The framing of the problem as “sequencing” aligns well with prior work in AI for education.

**Weaknesses:**

# Modification of PERM

- The proposed modification of PERM is not convincing.

- In the original PERM, the curriculum is designed so that the next task’s difficulty matches the learner’s current ability: $d_{t+1} \leftarrow a_t.$

- The paper proposes $d_{t+1} = \epsilon a_t$ for $\epsilon \geq 1.$

- This appears to be a simple rescaling. One could redefine ability as $a'_t = \frac{a_t}{\epsilon}$ and recover the original PERM formulation.

- I do not see how this models “faster learning rates” for humans. Without a stronger modeling argument, this feels like a cosmetic change.

- More importantly, the decision to let difficulty match ability in PERM comes from how the success probability is modeled. However the paper does not justify the contribution of having $d_{t+1} = \epsilon a_t$ with $\epsilon \neq 1$ with a similar argument. Even then, this appears as a trivial change.

# Task difficulty via PPO convergence

- Measuring task difficulty by PPO convergence time is not principled for human learners.

- PPO’s convergence depends heavily on hyperparameters and optimization dynamics.

- PPO may converge slowly on tasks trivial for humans, or quickly on tasks that are cognitively demanding.

- This metric may be fine for curriculum generation for PPO-like agents, but it is not a reliable proxy for human difficulty.

- The "convergence time ordering" may flip in tasks depending on the deep RL algorithm being used, the network architecture, and hyper-parameters. This makes this metric highly unstable and not reliable.

# Task similarity via Wasserstein distance

- I cannot find in the paper anywhere about the ground cost used in Wasserstein distance (i.e. what is the $d$ in equations 2 and 3 in Appendix A.2?). A Wasserstein distance is always with respect to a cost. What is $d(\phi_1, \phi_2)$? How is it defined?

- Without any information, I will assume this is the sum of weighted distances between state-action pairs from two trajectories, where distance between state-action pairs is either Euclidean in embedding space or Hamming for one-hot encoding.

- Using Wasserstein distance between state-action occupancy measures as a proxy for task similarity, especially with respect to knowledge transfer, is questionable for human learners.

- To give a counter-example: consider two levels of the same game with identical difficulty, except one has inverted controls (so UP takes you DOWN etc). Trajectory distributions for optimal policies would be far apart (large Wasserstein distance) because the mass over actions must be moved completely. Humans, however, adapt almost instantly once they recognize the inversion. **This will heavily depend on the choice of $d$ though.** You can try to pick a cost that makes Wasserstein distance label-invariant on the actions. But then for those we can come up with other counter-examples for tasks that have large differences in state-action occupancy measures yet are easily transferable for humans.


 # Post-training evaluation (Section 6.1)

- Results are not surprising, as comparisons are only against random and handcrafted curricula. In Jumper, handcrafted performs as well as PERM-H. In Emergency Response, no significant difference is observed between handcrafted and SimMAc. Matching expert performance could be a success, but the quality of handcrafted curricula is unclear since they were created by the research team. Without evidence that these are truly expert-designed, it is difficult to interpret the results.

# Comparison to prior work (Section 6.2)

- Results in Section 6.2 are thin. Figure 3 shows expected behavior: PERM-H starts with higher difficulty since it allows tasks above the learner’s ability. What is missing though is a direct comparison to standard PERM or other RL-based teaching approaches in the human experiments. Figures 1 and 2 should include such baselines.

- Earlier the paper states “no significant differences were found between SimMAC and handcrafted conditions,” but later claims “students’ better performance under the SimMAC curriculum indicates that…” These statements are inconsistent. SimMAc was not meaningfully better than handcrafted, and here we see that the cumulative distance of handcrafted actually matches random instead of SimMAc. So we cannot really draw any conclusions here.

- There needs to be at least one curricula generation approach that the authors compare to as baseline in teaching humans.  PERM and/or other RL-based teaching approaches should have been a baseline in human studies. Otherwise we don't know if this approach is better than prior work.

# Related work and references

- Several relevant works are not cited. I think (1) and (3) deserve a mention and is closely relevant. (2) is one of the earliest works about RL in teaching humans, which also deserves a reference in my opinion. (4) does not involve teaching humans but is methodologically very similar to the paper, so I think it deserves a mention as well.

(1) Mandel et al. (2014), Offline Policy Evaluation Across Representations with Applications to Educational Games

(2) Chi et al. (2011), Empirically Evaluating the Application of Reinforcement Learning to the Tutoring of Human Learners

(3) Singh et al. (2023), Learning to Learn: How to Continuously Teach Humans and Machines

(4) Raparthy et al. (2020), Generating Automatic Curricula via Self-Supervised Active Domain Randomization

- References need proper formatting with venues. For example, “Transferable Curricula through Difficulty Conditioned Generators” should be cited as IJCAI 2023. Given its closeness to this work (PERM), accurate citation is especially important.

**Questions:**

1. Can you provide a stronger modeling justification for PERM-H over PERM? What does this generalization allow you to model that PERM cannot, mathematically? Or can you prove that this model is strictly more general than PERM?

2. Why were no comparisons made against prior RL-based / curriculum-design based teaching approaches (e.g., PERM vs PERM-H) in the human experiments in terms of teaching performance?

3. Can you clarify the apparent contradiction between the statements about SimMAC vs. handcrafted curricula?

4. Based on the points I raised about using convergence time and Wasserstein distance as metrics for task difficulty and transferability, do you have any arguments / evidence that suggests these are indeed generalizable and accurate surrogates for teaching humans?

---

### Official Review · Reviewer_X4c9 · 2025-10-29

**Soundness:** 3
**Presentation:** 3
**Contribution:** 2
**Rating:** 4
**Confidence:** 4

**Summary:**

The paper proposes a practical warm-start for AI “teacher” algorithms for human learning, using data collected from RL agents in an exploration stage.  The authors introduce task diversity, using a generalization of a Partially Observed Markov Decision Process (POMDP) model, the Unspecified POMDP (UPOMDP).

**Strengths:**

The authors highlight important gaps in cold-start data necessary for AI teacher algorithms, namely that the necessary volume of human learning data for proper initialization of these models. This work’s primary contributions appear to be a “human-in-the-loop” style explore-exploit framework, by which the authors provide warm-start data for RL-informed AI teaching, and useful experimental results showing the relative strengths and weaknesses of RL-based data generation for warm starts (for these AI teaching agents). This explore-exploit formulation is an interesting formulation that fits nicely in the gaps identified by these authors. This paper’s proposal and experimental results are interesting, meaningful contribution to AI teaching algorithms. Particularly for SimMAC in the Emergency Response task, these experimental results demonstrate meaningful improvements in learning using Ai-teaching algorithms (with RL-informed warm-start)

**Weaknesses:**

- The paper acknowledges alignment gaps between RL agents and human learners but lacks a formal model or quantitative framework to analyze or predict such discrepancies.

- The novelty of the paper is unclear. The proposed PERM-H adapts an existing algorithm (PERM) with a simple scaling parameter (δ = εa), offering limited theoretical innovation. SimMAC extends existing similarity-based curriculum design (e.g., Li et al., 2023b; Silva & Costa, 2018) without introducing a clearly new learning formulation beyond combining difficulty and similarity metrics.

**Questions:**

- The novelty of PERM-H is unclear. From Section 4, it appears to be only a quite minor extension of the existing PERM algorithm’s optimal learning $\delta$. Is the novelty here the explore-exploit structure, within which the authors apply this algorithm? If so, I think this is understated when these stages are introduced in Section 4.

- As the authors note in the Discussion, in certain tasks (e.g. the Jumper activity in this paper), RL-based learning may require substantial training to mirror human learning. This is not necessarily a fault of the work but does somewhat contradict some claims in the Introduction, that these human-in-the-loop methods can be time/compute savers or substitutes for current workflows. In particular, the authors could reckon claims like

    *“However, gathering comprehensive human learning data is time-consuming and costly; in one study, it took approximately 900 man-hours for a Machine Learning-based teacher algorithm to converge (Bassen et al., 2020).”*

    as a motivation of their method with

    *“In the Jumper environment, we corroborate the results of Tsividis et al. (2017), with humans demonstrated superior learning efficiency, reaching high performance levels quickly while RL agents required millions of experiences to achieve even minimal human performance levels. Despite this gap, RL agents and humans showed consistent agreement on task difficulty rankings. The alignment suggests that in carefully designed domains, RL can effectively provide valid initial training data in place of human learners”*

- Further details on the parameters (i.e. “task pool”) of the Emergency Response task would be helpful. I may have missed these details but did not see them clearly in the Appendix summary. Figure 16’s flowchart does not appear to be comprehensive or quantify/characterize this complexity, and it appears important to include additional task diversity compared to Jumper’s two parameters.

- How was “convergence” defined/evaluated for tasks? Was this criteria task-specific or a general criteria used across tasks?
- Can the authors comment on the creation of the “handcrafted” training materials? The statement in equation 1 is not extremely informative for how these tasks were crafted. Are there other tasks with “silver/gold”-standard hand-crafted curricula to which these algorithms (e.g. SimMAC) could be compared? It is unclear if this was an effective curricula, other than in improvement over randomly assigned “levels” or otherwise no “playing”

---

### Official Review · Reviewer_Sptr · 2025-10-30

**Soundness:** 2
**Presentation:** 3
**Contribution:** 2
**Rating:** 2
**Confidence:** 3

**Summary:**

- This paper studies the problem of automatically developing teaching curricula for human learners
- They describe a two-stage framework for generating this teaching curricula. (1) Using synthetic RL agents to simulate human learning progress. (2) Training a model on this synthetic learning data to maximize efficient learning
- They describe two algorithms for training a teacher model - PERM-H, a modification of the prior PERM algorithm, and SimMAC, a novel algorithm which leverages task difficulty and knowledge transfer to generate the task ordering.
- They introduce two new environments for student learning, a platformer game, and a emergency medical-care simulator.
- They perform human-subject experiments evaluating PERM-H in the platformer game and evaluating SimMAC in the emergency response game, demonstrating that these approaches outperform a random baseline and match handcrafted curricula that they created.

**Strengths:**

- This paper discusses a very important and interesting application - automatically generating learning curricula for human learners
- The human subject experiments in multiple domains show promise that this approach could work and generalize to real human learners
- The paper is overall very clear and directed, and the narrative of using synthetic agents to simulate humans for learning is present throughout, making the paper easy to follow

**Weaknesses:**

- PERM-H
    - The authors list PERM-H as one of the key contributions of this work, but there are many details missing here. The only change between PERM-H and PERM is the introduction of the ϵ parameter, but there is no discussion on how we can select this ϵ, any intuitive explanation of what this parameter means, or even which ϵ was used in the results. The figures in the main paper only show results for PERM-H, and in the appendix, PERM and PERM-H are used seemingly interchangeably (e.g. in Figures 6/7/8, the plots mention PERM while the captions mention PERM-H, leading to further confusion.
- Handcrafted
    - In both domains, the authors designed a “Handcrafted” learning curricula to use as a baseline, and described them as easy-to-hard learning curricula. However, there was no other description of how the handcrafted curricula was generated, and considering both domains are completely novel as well, using this underspecified handcrafted curricula as a core baseline for evaluation is slightly dubious.
- RL Agents
    - There is little information on how the RL students were trained. Lines 154-157 mention that RL agents are used to simulate student interactions, but there are no details on which RL algorithms were used or experiment parameters. The authors use a PPO-based student to evaluate the final trained teacher model (Line 428, 687) in Jumper, so I assume the initial students used to generate the training dataset were also PPO, but this is not explicitly stated. There is no model description I could find for Emergency Response.
    - The authors repeatedly mention that these RL-agents and the datasets they generate are diverse, but there is no elaboration on how this diversity was induced, or analysis of this diversity on the final dataset. The appendix mentions a RL-agent training dataset size of 14506 for Jumper, but there is no discussion on how much is really necessary to beat the random baseline, how the results are affected by dataset size and diversity.
    - The core narrative of this paper is that the authors aim to simulate human learning using RL agent learning, but there is little discussion on how human-like these RL agents behave, or whether the RL learning process itself is human-like. I am not entirely satisfied by the brief analysis done in Section 6.3, considering how important this concept is to the overall paper. In particular, I’m not sure that Figure 3 justifies the statement that “RL agents and humans showed consistent agreement on task difficulty ratings”. Figure 5 is slightly more compelling, but is lacking any description on which RL agents were used to generate these plots? Are these untrained agents, agents selected during training, converged agents, or something else?
- Comparison with prior work
    - In the related work section, the authors mention that prior works failed to beat baselines across domains. However, the proposed approaches are only evaluated in two entirely new domains, and not in domains that previous works failed in.
- Minor points
    - On Line 869, you mention that your results contrast with typical RL agent approaches, but this is not a fair comparison in my view - the question of dataset ordering and dataset composition are orthogonal questions, and there is a rich body of work talking about curriculum learning approaches in RL
  - The paper is missing a reproducibility statement. The authors did release the full code for Jumper in the supplementary materials, but not the Emergency Response domain.
  - The paper is missing an ethics statement. Though I didn't see any specific ethical issues in this work, this statement would be helpful considering there was a human subjects study involved in the work.
- Overall conclusion
    - Overall, I think this work investigates a very important problem, and suggests a natural solution to the challenge of limited human data. However, there is very limited theoretical analysis to justify the use of RL agents as appropriate stand-ins for human learners, and the empirical results are not strong enough to justify this approach (beating a random baseline and an underspecified first-party, handcrafted baseline).

**Questions:**

- Mostly, I am looking for more details many of the sections which I mentioned in the weaknesses section (PERM-H, RL Agent training, how the handcrafted curricula were generated, more rigorous comparisons between RL Agents and Humans).
- Could you situate the SimMAC approach better in the literature? Have other SimMAC-like approaches leveraging the zone of proximal development been evaluated deployed in practice with human learners? (possibly in data-rich domains or platforms like Duolingo or IXL).

---

### Note · Authors · 2025-11-18

**Comment:**

We thank all our reviewers for their comments and their time reviewing their work. We are glad that the reviewers acknowledged the value and the importance of our work. Based on the current feedback, we have decided to withdraw from ICLR 2026; our intention for this work was meant to investigate and demonstrate the viability of RL as a warm-start for human learning, something which we believe has not been done before. Thus, our intended focus for this work was the 2-step framework and our human subjects study. This led to criticism that our proposed algorithms were not novel over previous work, which we accept. We could have written the paper in a better way to focus on the framework and our response to the challenge of the cold-start problem.

Here are some responses to the queries made by the reviewers:

1. On the handcrafted condition design:
   1. Jumper: There are two parameters, which is *spike\_density* and *ground\_roughness* which is represented by a Bernoulli and Categorical Distribution respectively. During level generation, each block of the level will sample from these values to determine if there is a spike at the current unit, and the height of the ground, for a total of 48 blocks to form the level. Across the 10 training levels, spike\_density increases at a linear schedule from 0.0 to 1.0, while ground\_roughness increases in entropy (e.g. \[0.0,1.0,0.0\] → \[0.33,0.33,0.33\]).
   2. Emergency Response: There are a total of 10 medical conditions, which we use in combination to create the tasks, and use to verify the viability of task similarity in our work. The difficulty is determined by the complexity of the underlying condition, which directly correlates with the depth of the flow chart in Figure 16\. For the handcrafted curriculum, participants will undergo all medical conditions in increasing order of difficulty(e.g. heatstroke\_easy, heatstroke\_hard, seizure\_easy…). The order of the presentation of the conditions were randomized and fixed for all participants.
   3. In our paper, we have shown that it is still a valid curriculum from the human subjects studies, beating our other baselines. We have shared the parameters and design here for transparency. This will also be included in all future manuscripts.

2. On prior work and comparisons in studies:
   As we have covered in the related works section, there are constraints in implementing prior works in UED and Curriculum Design, as they were designed with RL agents in mind, and not humans, necessitating different considerations. For example, one could easily imagine that training humans with ACCEL within 10 levels is akin to a random curriculum. To our knowledge, no algorithm, other than Domain Randomization, has shown to be practically feasible with both RL agent and Humans.

   We selected both handcrafted and random curriculum because it's well-studied in the pedagogy literature. While our domains are new, both baselines are still very robust across different domains in human learning, despite its simplicity. Thus, we reject the comment that the results are “not surprising", since a large proportion of prior works with human subjects fail to beat a random curriculum, largely owing to the Spaced Repetition effect.

3. As stated in our README, the Emergency Response environment was too big for OpenReview, but is prepared for open release.

**Withdrawal Confirmation:**

I have read and agree with the venue's withdrawal policy on behalf of myself and my co-authors.